# The Presence of Mycotoxins in Human Amniotic Fluid

**DOI:** 10.3390/toxins13060409

**Published:** 2021-06-09

**Authors:** Karolina Gromadzka, Jakub Pankiewicz, Monika Beszterda, Magdalena Paczkowska, Beata Nowakowska, Rafał Kocyłowski

**Affiliations:** 1Department of Chemistry, Poznań University of Life Sciences, ul. Wojska Polskiego 75, 60-625 Poznań, Poland; 2Premedicare, NEW MED Medical Center, ul. Drużbickiego 13, 60-693 Poznań, Poland; biuro@new.med.pl (J.P.); rkocylow@gmail.com (R.K.); 3Department of Food Biochemistry and Analysis, Poznań University of Life Sciences, ul. Mazowiecka 48, 60-623 Poznań, Poland; monika.beszterda@up.poznan.pl; 4The Institute of Mother and Child, ul. Kasprzaka 17a, 01-211 Warszawa, Poland; magdalena.paczkowska@imid.med.pl (M.P.); beata.nowakowska@imid.med.pl (B.N.)

**Keywords:** mycotoxins, *Aspergillus*, *Fusarium*, amniotic fluids, fetal defects, genetic abnormalities

## Abstract

Mycotoxin exposure assessments through biomonitoring studies, based on the analysis of amniotic fluid, provides useful information about potential exposure of mothers and fetuses to ubiquitous toxic metabolites that are routinely found in food and the environment. In this study, amniotic fluid samples (n = 86) were collected via abdominal amniocentesis at 15–22 weeks of gestation from pregnant women with a high risk of chromosomal anomalies or genetic fetal defects detected during 1st trimester prenatal screening. These samples were analyzed for the presence of the most typical *Aspergillus*, *Penicillium* and *Fusarium* mycotoxins, with a focus on aflatoxins, ochratoxins and trichothecenes, using the LC-FLD/DAD method. The results showed that the toxin was present in over 75% of all the tested samples and in 73% of amniotic fluid samples from fetuses with genetic defects. The most frequently identified toxins were nivalenol (33.7%) ranging from <LOQ to 4037.6 ng/mL, and aflatoxins (31.4%), including aflatoxin G1, ranging from <LOQ to 0.4 ng/mL. Ochratoxin A and deoxynivalenol were identified in 26.7% and 27.9% of samples, respectively. Bearing in mind the above, the detection of mycotoxin levels in amniotic fluid is useful for the estimation of overall risk characterization with an attempt to link the occurrence of fetal abnormalities with exposure to mycotoxins in utero.

## 1. Introduction

To date, it is believed that the pregnancy and the good health of newborns are particularly influenced by maternal infections, hypertensive disease in pregnancy, smoking and insufficient maternal nutrition [1,2]. Moreover, environmental pollution has recently been added to this list of threats to the health of the fetus. In addition, food consumed by pregnant women may contain dangerous natural products such as mycotoxins.

Mycotoxins are hazardous metabolites which are produced by fungi. Those fungi are divided into two groups: field fungi that invade before harvest (mainly *Fusarium* species), and storage fungi that occur only after harvest (*Aspergillus* and *Penicillium* species). Mycotoxins show harmful effects on humans, animals and plants, which result in diseases and economic losses. Worldwide feed and food contamination with those secondary metabolites is a significant problem [3,4,5].

Food can be contaminated with toxins at numerous stages in the food chain [3] and *Aspergillus*, *Alternaria*, *Claviceps*, *Fusarium*, *Penicillium* and *Stachybotrys* are the most common genera of mycotoxigenic fungi. So far, more than 400 mycotoxins have been identified. Scientific attention is mainly focused first of all on carcinogens and/or metabolites that are well known as being toxic to humans and animals. Aflatoxins (aflatoxin B1 (AFLB1)), ochratoxins (ochratoxin A (OTA)), fumonisins (fumonisin B_1_ (FB_1_)), zearalenone (ZEA) and trichothecenes (deoxynivalenol (DON)) are considered to be the most significant mycotoxins in agriculture and in the food industry [6]. These substances are toxic to vertebrates and humans at low concentrations, with various acute and chronic effects, depending on the species. Analyzing a given species, the effect of mycotoxins on health is influenced by age, sex, diet, weight, the presence of other toxic metabolites (synergistic and antagonistic effects) and pharmacologically active substances, and also by exposure to infectious agents [5,7]. Most of the mycotoxins known so far have been grouped according to their toxic properties, under chronic conditions, as teratogenic, mutagenic or carcinogenic. They can also be grouped according to their target place of action, which results in the designations of neuro-, hemo-, nephron-, hepato-, dermato- and immunotoxins [8]. Human exposure to mycotoxins can result from the consumption of food contaminated with toxins (mainly cereal food), from the transfer of mycotoxins and their metabolites in animal products like meat and eggs or from exposure to air containing dust and toxins [4,5].

To prove that a disturbance is due to mycotoxicosis, a dose–response relationship between the substance and the disease must be demonstrated. In the case of human populations, this dependence requires epidemiological studies. Additional evidence is provided when the typical symptoms of suspected mycotoxicosis in humans are induced reproducibly in animal models through exposure to a given mycotoxin. Human exposure to mycotoxins is further determined by environmental or biological monitoring. In environmental monitoring, metabolites of toxigenic fungi are measured in air, food or other samples. In biological monitoring, the presence of mycotoxins, adducts and residues is tested directly in tissues, fluids and excreta [9].

So far, there is no information about how toxins found in food affect the health of pregnant women and the condition of the fetus. The available work focuses on aflatoxins and their content in blood. The assessment of the concentration of aflatoxins in the maternal and cord blood and its influence on birth weight has been performed in various countries and populations [6,7,10,11,12,13,14,15,16]. However, not much is known about the presence of mycotoxins in the amniotic fluid [10]. The composition of the amniotic fluid (AF) is relevant to the development of the fetus and reflects both fetal and maternal compartments. During pregnancy, physiological and metabolic changes in the organs of a woman depend on the requirements of the growing fetus. The composition of AF is influenced by the mother’s diet, including essential nutrients, and the exposure scale to toxic substances. Overexposure to toxic substances can be harmful to the health of both the fetus and the pregnant woman [11,12].

The fetus has a unique sensitivity to deleterious health effects resulting from in utero exposure to environmental pollutions, because of rapid cell proliferation, poor detoxification or elimination capacity, high absorption rate and immature repairing mechanisms [13]. Human epidemiological studies have indicated that in utero exposure to xenobiotics could contribute to the development of cancer in early life, or can even predispose the fetus to chronic disease and carcinogenesis later in life. In the case of the excretion of xenobiotics to fetal urine, this ends up in the amniotic fluid, which is then subject to subsequent fetal swallowing. Considering the genotoxic, neurotoxic, nephrotoxic and endocrine-disrupting properties of fungal metabolites, early exposure and continuous circulation of these substances in the fetus throughout pregnancy could interfere with critical fetal development [13].

Based on the currently available experimental results, it is probable that transplacental transfer of mycotoxins—metabolites of low molecular weight—occurs from the early gestational stage and accumulates in the fetal circulation towards the end of pregnancy. Taking the above into account, the major aim of this study was the analysis of the presence of mycotoxins in amniotic fluids taken from women with abnormalities in the development of the fetus, including genetic defects.

## 2. Results and Discussion

In this study, 86 samples of amniotic fluids were analyzed for the presence of mycotoxins. The toxins that formed in the field (zearalenone, deoxynivalenol and nivalenol) and those created during the storage of grain products (ochratoxin A and aflatoxins B1, B2, G1 and G2) were analyzed. Toxins were selected for research on the basis of previous studies and their frequency and harmfulness to humans and animals. Deoxynivalenol and nivalenol belong to the trichothecenes. These compounds are toxic to humans and animals. They cause hemorrhages and can lead to cardiac arrest and death [3,4,9]. Zearalenone is a lactone of resorcinic acid—this compound is classified as phytoestrogenic, which in practice means it has a special effect on human and animal reproduction. The effect of zearalenone is several times stronger than that of natural estrogen [3]. Aflatoxins and ochratoxin A belong to the storage mycotoxins, i.e., they are formed during improper storage. The first is mutagenic, teratogenic and hepatotoxic [10]. On the other hand, ochratoxin A belongs to the group of ochratoxins with mutagenic activity, and also has nephro-, neuro- and immunotoxic properties [13].

Amniotic fluid samples were collected between weeks 15 and 22 of gestation from women suspected for fetal genetic abnormalities during first trimester screening (between 11 and 14 weeks). Among the tested samples of amniotic fluid, 71 came from fetuses with no genetic defects, whereas 15 came from women whose offspring were confirmed to have genetic disorders. The conducted research showed that the toxin was present in over 74% of the tested samples and in 73% of amniotic fluid samples from fetuses with chromosomal aberrations (Table 1).

The available literature generally lacks information on the occurrence of mycotoxins in amniotic fluid. However, one can find information about the presence of these compounds in the blood of the mother or the cord blood. These data mainly refer to storage mycotoxins, especially aflatoxins [10,11,12,13,14,15,16].

According to Kyei et al. [17] there is already some evidence to suggest that exposure to toxic metabolites of fungi during pregnancy could have deleterious effects on pregnancy outcomes. However, given the slight number of studies, especially on effects of *Fusarium* toxins, more studies are necessary for a more comprehensive understanding of the effects of various mycotoxins on maternal and fetal health condition and to inform public health policies and interventions.

During our research, aflatoxins were detected in 31.4% of all analyzed samples. The most frequently aflatoxin was G1, which was present in 17.4% of the samples at concentrations ranging from <LOQ to 0.4 ng/mL. Aflatoxins G2 and B2 were present in 10.5% of the samples, with maximum concentrations of 1.9 and 0.4 ng/mL, respectively. Aflatoxin B1 was detected in only one analyzed sample (1.2%) at a concentration of 0.2 ng/mL. The results show that among the amniotic fluid samples in which the aflatoxin was present, genetic defects were found in 22.2% of fetuses (Table 1 and Table 2; Appendix A). Thus, no relationship was found between the presence of aflatoxins and the occurrence of genetic defects in the fetus.

The first studies on the aflatoxin content in the mother’s blood and in the cord blood were carried out in 1989 by De Vries et al. [18]. According to the authors, aflatoxins (AFLs) were detected in concentrations ranging from 12 to 11,574 pg/mL in 53% of maternal blood samples, and in concentrations ranging from 17 to 6819 pg/mL in 37% of cord blood samples. There was a relationship between sex and AFL content in maternal blood at delivery on birth weight because in AFL-positive mothers the mean birth weight of female infants was about 255 g lower than that of females born to AFL-negative mothers. An inverse dependence was observed in the case of male infants, which means the birth weights of males born to AFL-negative mothers was 132 g lower than that of males born to AFL-positive mothers.

According to the literature, aflatoxins are present in the cord blood of 38% to 91% of tested samples at concentrations of 4–10,440 pg/mL. There is also no clear information on the impact of the presence of toxins on the birth weight [17,19,20,21,22,23,24]. As for maternal blood, this type of toxin was detected in 53% and 68% of the tested samples at concentrations of 12–11,574 pg/mL [18,20]. According to the latest research, aflatoxin was found in all blood samples taken from pregnant women and their content ranged from 0.44 to 268.7 pg/mL [16]. The amount of aflatoxins in the mother’s blood will certainly depend on the geographic region and diet.

Most of the work on the presence of mycotoxins and their impact on the health conditions of fetuses and newborns refer to the presence of these compounds in the mother’s blood and cord blood. Therefore, comprehensive research is extremely important in order to draw the right conclusions. Ibeh et al. [10] also examined the content of toxins in amniotic fluids. Aflatoxin values above 20 ppb were obtained in 74.1% of samples of the amniotic fluid; 67.2% of neonatal cord blood samples and 62.4% of venous maternal blood samples. According to the authors, these results suggest that these mycotoxins present in the maternal blood cross the placental barrier and can accumulate in the fetus, which further explains the high amount of aflatoxins in the amniotic fluid and the intrauterine exposure to toxins.

In addition to aflatoxins, Jonsyn et al. [19] also studied ochratoxin A concentrations in cord blood. This toxin was present in 25% of the tested samples at concentrations of 200–3500 pg/mL. Exposure to ochratoxin A (OTA) had no effect on boys’ birth weights. The average birth weights of girls exposed to the toxin were 190 g lower than those of girls who were not exposed. In the Czech Republic, OTA was detected in almost all tested serum samples (n = 115) from pregnant women in amounts up to 1.13 µg/L, and in women of the child-rearing age (n = 100) up to 0.35 µg/L [25,26]. Simultaneously, these data were correlated with OTA dietary exposure assessments. OTA has also been identified in serum samples (n = 98) of pregnant women from Egypt [13], where it was in the range of 0.20–1.53 ng/mL and the estimated OTA concentration in fetuses was 0.40–3.06 ng/mL. In turn, Ritieni et al. [14] examined ochratoxin in amniotic fluid. OTA was present in only one of 21 tested samples in a concentration of 4.26 ng/mL. In our studies, OTA was found to be present in 26.7% of examined samples. The toxin concentration was in the range of <LOQ to 0.7 ng/mL. Figure 1a shows all the samples in which the amount of the toxin could be analyzed (16 samples). In the remaining cases, the OTA concentration was below the quantitative limit. Only three OTA samples were found in the amniotic fluid of the fetuses which had genetic defects (Table 1). Although the presence of toxins clearly does not affect the occurrence of genetic defects, it may have an impact on the health of the fetus or its birth weight.

The presence of OTA in the amniotic fluid is in agreement with the literature, where the carry-over of the mycotoxin through the placenta has been reported [27,28].

During this study, the toxins metabolized by the *Fusarium* species were also analyzed. There is no information in the literature on the presence of deoxynivalenol, nivalenol (NIV) and zearalenone in amniotic fluids and their effects on the fetus. The chromatographic analysis showed that zearalenone was not present in all 86 samples of amniotic fluid. Our hypothesis is that this toxin does not cross the placenta; however, there is no information in the literature about the presence of this toxin in the cord and maternal blood. Therefore, further studies are needed to confirm that this toxin does not pose a risk to the fetus.

The most commonly detected toxin in the amniotic fluid was nivalenol. It was present in 33.7% of the tested samples at concentrations from <LOQ (1 sample) to 4037.6 ng/mL (Table 1; Figure 1b). The presence of the toxin in such large amounts is disturbing, although in the case of this toxin there was also no relationship between its presence and the occurrence of genetic defects. The frequent occurrence of nivalenol in pregnant women in Poland is justified because NIV is one of the most common mycotoxins in our country. NIV was present in 29 tested samples. Of these, four fetuses had genetic defects (13.8%). In animal studies NIV shows higher acute toxicity than DON, with oral LD_50_ values in mice of 39 and 78 mg·kg^−1^ for NIV and DON, respectively [29]. At the molecular level, NIV and DON demonstrate multidirectional inhibitory effects on the primary metabolism of eukaryotic cells, including the inhibition of DNA, RNA and protein synthesis. Such damage leads to disturbed cell proliferation in tissues with high rates of cell turnover such as bone marrow, intestinal mucosa, spleen and thymus [30,31,32]. As demonstrated by Cheat et al. [30], NIV has a stronger influence on the intestinal mucosa than DON (NIV enhanced apoptosis at the top of villi and decreased the proliferative/apoptotic cell ratio by about half), both in vitro and in vivo, which highlights the inevitability of a specific hazard characterization for NIV risk assessments.

The second trichothecenes toxin analyzed in the amniotic fluid was deoxynivalenol. It was present in 24 of 86 tested samples (Table 1; Figure 1c). The DON content in the amniotic fluids ranged from <LOQ (1 sample) to 2417.8 ng/mL. Only three samples in which DON was found were from women who gave birth to a child with a genetic defect. There is a lack of information on the presence of deoxynivalenol in tissues, fluids and excreta, and on its potential impact on the health of the mother and the fetus.

So far, the content of DON has been analyzed only in the urine of pregnant women [33]. Total DON (free-form and DON glucuronide combined) was identified in 88.1% and 83.3% of women on days 1 and 2, respectively. On day 1 the mean concentration of total DON was 29.7 (in the range from 0 to 436), and on day 2 it was 28.7 ng/mL urine (range 0–167). The high incidence of the presence of DON in the urine of pregnant females from the UK in this study confirms its ubiquitous presence in cereal food products. According to Rodríguez-Carrasco and coworkers [34], after 49.2 µg of DON daily intake, a urinary daily excretion of 35.2 µg was identified in humans, representing 68.3% of the established DON provisional maximum tolerable daily intake (PMTDI).

One of limitations of our study is the use of human amniotic fluid samples from high-risk pregnancies only. Such samples from normal healthy pregnancies could be collected, only if the woman agrees to amniocentesis. Due to the invasive and therefore risky character of AC, informed consent is very difficult to obtain when not medically indicated. However, samples with normal genetic results may constitute a kind of control subgroup for genetically abnormal samples in the high-risk population.

## 3. Conclusions

The limited developmental toxicology literature implies possible adverse effects of mycotoxins on fetal growth, viability and postnatal development. In our study, amniotic fluid samples collected in the second trimester (15–22 weeks of gestation) via amniocentesis revealed the presence of mycotoxins in more than 75% of the tested samples and in 73% of amniotic fluid samples from fetuses with genetic defects. The most frequently identified toxin was nivalenol, which is simultaneously one of the most common mycotoxins contaminating crops in Poland. Moreover, the presence of aflatoxins ochratoxin A and deoxynivalenol were also observed in 26.7% to more than 31% of samples. The detection of several mycotoxins in amniotic fluid indicates that, in addition to blood borne in utero exposure, the fetus is also ingesting and bathing in low levels of these xenobiotics, which may exacerbate the risk of developing chromosomal anomalies or genetic fetal defects.

## 4. Materials and Methods

### 4.1. Chemicals and Reagents

Mycotoxin standards (AFLG1, AFLG2, AFLB1, AFLB2, OTA, DON, NIV, ZEA), HPLC-grade solvents and all reagents for extraction and purification process were obtained from Sigma-Aldrich (Steinheil, Germany). Water (HPLC grade) was obtained from Milli-Q systems (Millipore, Billerica, MA, USA).

### 4.2. Study Design

This was an exploratory study to determine the levels of mycotoxins in AF in the second trimester, between 15 and 22 weeks of gestation. The AF samples were collected during routine diagnostic amniocentesis in pregnant women with a high risk of chromosomal abnormalities.

### 4.3. Biospecimen Collection and Preparation of Amniotic Fluid

Amniotic fluid (AF) samples were obtained (15–20 mL) for genetic analysis following routine diagnostic amniocentesis at 15–22 weeks of gestation. Samples were collected by means of transabdominal aspiration using sterile 22–25 G spinal and diagnostic puncture needles and plastic syringes that were free from organic and non-biological contamination. Each sample of the fresh amniotic fluid was centrifuged for 5 min at 1300 rpm. Next, 2 mL of supernatant was transferred to a new 2 mL tube and frozen at −20 °C for further mycotoxin analyses.

Genomic DNA for chromosomal microarray analysis (CMA) was extracted from the fresh amniotic fluids (AFs) using the Sherlock kit (A&A Biotechnology, Poland) according to the manufacturer’s recommendations. Array comparative genomic hybridization (array CGH) was performed using 60K microarrays 8 × 60K from Oxford Gene Technology (CytoSure ISCA, v3). The array contained 51,317-mer oligonucleotide probes, covering the whole genome, with an average spatial resolution of 60 Kb.

Procedures for DNA denaturation, labeling, hybridization and washing were performed according to the manufacturer’s instructions. Genomic DNA was labeled using a CytoSure Labeling Kit (Oxford Gene Technology, Begbroke, United Kingdom), with no enzyme digestion. Hybridization was performed between 24 to 48 h at 65 °C in a rotator oven (Agilent, Santa Clara, CA, USA). Arrays were washed using Agilent wash buffer 1 and 2, scanned using an Agilent Technologies microarray scanner and the signal intensities were measured using Feature Extraction software (Agilent Technologies, CA, USA). Agilent Feature Extraction software (V10.0) was used to quantify all scanned images.

CytoSure Interpret Software (Oxford Gene Technology, Begbroke, UK) was used to perform data analysis based on the reference genome (NCBI37/hg19).

In most cases, part of the amniotic fluid sent for aCGH was also used for cytogenetic karyotype analysis. Amniocytes were cultured using in situ vessels. Colony growth and mitotic activity were controlled daily after the 8th day of culture. Metaphase chromosomes were prepared according to the standard procedures of GTG banding.

### 4.4. Mycotoxins Determination

An aliquot of 500 microliters of each sample (amniotic liquid) was added of 2.5 mL of a 0.1 M MgCl_2_ solution and the mixture was stirred; subsequently, 2.5 mL of chloroform were added, the solution was acidified with HCl 6N and the mixture was stirred again for 2 min. The samples were then ice-cooled for 20 min and then centrifuged at 4 °C for 10 min at 3000 rpm. The organic phase was separated and the organic solvent was evaporated using a ThermoSavant centrifugal evaporator (Savant Instruments Inc., Farmingdale, NY, USA). Before the chromatographic analysis the samples were dissolved in 200 microliters of a solution of mobile phase.

The toxin content was determined using the chromatographic system: a Waters 2695 high performance liquid chromatograph, a Waters 2475 Multi λ Fluorescence Detector and/or a Waters 2996 Array Detector. The quantification limits (LOQs) were determined by multiplying the detection limits (LODs) by 3.3. The LODs of the methods were calculated using a signal-to-noise ratio of 3:1.

Zearalenone determination was carried out using a fluorescence detector and the excitation and emission wavelengths were 274 and 440 nm, respectively. The reserve-phase column was a C-18 Nova Pak column (3.9 × 150 mm), whereas the mobile phase was acetonitrile:water:methanol (46:46:8, *v*/*v*/*v*), at a flow rate of 0.5 mL/min. Quantification of ZEA was performed by measuring the peak areas at the ZEA retention time according to the relevant calibration curve (correlation coefficient R = 0.9998). The limit of ZEA qualification was 0.06 ng/μL. In order to confirm the presence of zearalenone, a photodiode array detector was used.

For deoxynivalenol and nivalenol determination, the trichothecenes were quantified by means of the HPLC method using a C-18 Nova Pak column (3.9 × 300 mm) and a Waters 2996 Array Detector (λmax = 224 nm for DON and NIV). DON and NIV were eluted from the column with a 25% water solution of methanol (flow rate 0.7 mL min^−1^) with retention times of 11.72 and 7.46 min, respectively. The detection limit for DON and NIV was 0.01 μg/mL. The quantification limit was 0.033 μg/mL. Positive results (on the basis of retention times) were confirmed by means of HPLC analysis and through comparisons with the relevant calibration curve (correlation coefficients for NIV and DON are 0.9994 and 0.9997, respectively).

The determination of OTA was carried out in a high-performance liquid chromatography (HPLC) system equipped with a fluorescence detector. Liquid chromatography was performed on a C-18 Nova Pak column (3.9 × 150 mm) column, operated at 0.9 mL/min with acetic acid:acetonitrile:water (2:99:99) as the mobile phase. The excitation wavelength of the fluorescence detector was set at 333 nm and the emission wavelength was 477 nm. For the quantitative determination of OTA, peak areas of the samples were correlated with the concentrations according the calibration curve. The qualification limit for OTA determination was 0.02 ng/mL.

Aflatoxin (G1, G2, B1, B2) quantities of samples were determined using HPLC with fluorescent detection and a C-18 Nova Pak column (3.9 × 150 mm). Aflatoxins were separated in the HPLC column with a mobile phase of water:methanol:acetonitrile (60:30:15) at a flow rate 1.2 mL/min. Fluorescence detection was performed at an excitation wavelength of 365 nm and an emission wavelength of 440 nm. Positive results (on the basis of retention times) were confirmed via comparisons with the relevant calibration curve. The qualification limit for aflatoxins G1 and G2 was 0.02 ng/mL and for B1 and B2 it was 0.03 ng/mL.

### 4.5. Ethics Approval and Consent to Participate

The study protocol was approved by the Bioethics Commission at Poznan University of Medical Sciences (approval no. 297/17, from 2 March 2017). Informed consent was obtained from all women. The study was performed in accordance with the Helsinki Declaration.

## Figures and Tables

**Figure 1 toxins-13-00409-f001:**
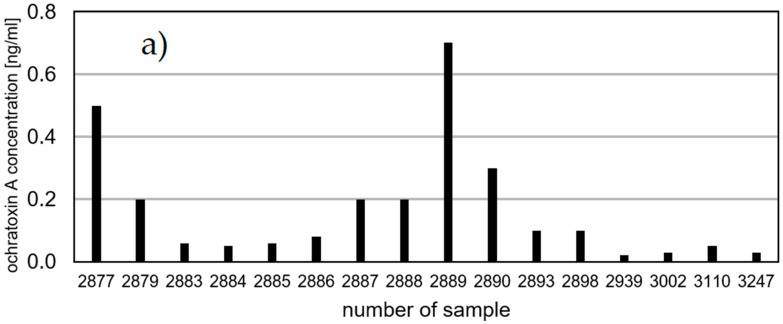
Mycotoxin concentrations in amniotic fluid (ng/mL)—(**a**) ochratoxin A, (**b**) nivalenol, (**c**) deoxynivalenol.

**Table 1 toxins-13-00409-t001:** The research data and summary of the presence of mycotoxins in human amniotic fluid samples with normal and abnormal genetic results.

Number of Samples Tested	Normal Genetic Samples	Chromosomal Aberration Samples	All Samples
	71	15	86
Toxins in amniotic fluid	Normal genetic samples (%)	Chromosomal aberration samples (%)	All samples (%)
All toxins	74.6	73.3	74.4
Zearaleone	0	0	0
Deoxynivalenol	29.6	20.0	27.9
Nivalenol	28.2	26.6	27.9
Ochratoxin A	28.2	20.0	26.7
Aflatoxin B1	1.4	0	1.2
Aflatoxin B2	9.8	13.3	10.5
Aflatoxin G1	16.9	20.0	17.4
Aflatoxin G2	9.8	13.3	10.5

**Table 2 toxins-13-00409-t002:** Aflatoxin concentration in positive samples of amniotic fluids.

nr	Nr PD	Test Results aCGH	AFLs [ng/mL]			
			G1	G2	B2	B1
1	2883	incorrect	nd	nd	<LOQ	nd
2	2887	correct	nd	nd	nd	<LOQ
3	2888	incorrect	nd	0.9	nd	nd
4	2902	correct	0.3	nd	nd	nd
5	2904	incorrect	<LOQ	<LOQ	nd	nd
6	2906	correct	<LOQ	<LOQ	nd	nd
7	2907	incorrect	<LOQ	nd	nd	nd
8	2920	correct	<LOQ	<LOQ	<LOQ	nd
9	2921	incorrect	nd	nd	0.4	nd
10	2930	correct	<LOQ	nd	nd	nd
11	2933	correct	nd	<LOQ	nd	nd
12	2936	correct	nd	nd	<LOQ	nd
13	2940	correct	nd	<LOQ	nd	nd
14	2941	correct	<LOQ	<LOQ	<LOQ	nd
15	2947	correct	<LOQ	nd	nd	nd
16	2949	incorrect	0.3	nd	nd	nd
17	2950	correct	0.2	nd	nd	nd
18	2951	correct	nd	<LOQ	nd	nd
19	2953	correct	nd	nd	<LOQ	nd
20	2954	correct	0.2	nd	nd	nd
21	2966	correct	nd	<LOQ	nd	nd
22	2994	correct	0.4	nd	nd	nd
23	3094	correct	0.02	nd	nd	nd
24	3113	correct	nd	nd	0.1	nd
25	3114	correct	nd	nd	0.2	nd
26	3115	correct	0.1	nd	0.2	nd
27	3227	correct	0.1	nd	nd	nd

## Data Availability

Not applicable.

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
