# Peer review of "The Presence of Mycotoxins in Human Amniotic Fluid"

_toxins, 2021, doi:10.3390/toxins13060409_

Round 1
Reviewer 1 Report
The authors described some interesting results on the presence of mycotoxins in human amniotic fluids. Major concerns:
- There are several main pieces in the methods part missing. For example, several aflatoxins are reported in the manuscript, are they all obtained from Sigma-Aldrich? How many total? Are they all use the same detection method (i.e., fluorescence, with the same excitation and emission wavelength)?
- What is the Limit of Quantitation (LOQ) for each analyte? How did they determine the LOQ?
- Clarify the mycotoxin family regarding the toxins they detected, for example, DON and NIV belong to trichothecene, OTA and ZEA belong to OCAs.
- All samples authors used are from those with high risk for chromosomal abnormalities. It is interesting to see and compare the results from normal pregnant women. At least should discuss that in the manuscript in light of the authors' findings that the presence of mycotoxins in those high-risk population.
- Table 1 is one of the main results and is listed in the supplementary materials. Authors should summarize those results for AFs into the Figures with different AFs in the main text to be consistent with other mycotoxins presented in the manuscript. Authors should also list the quantitative results from other mycotoxins in the supplementary materials.
Minor issues:
The authors need to carefully proofread the manuscript, for example, line 264, should be 500 microliters rather than micron.
Author Response
Reviewer 1.
The authors thank you very much for valuable tips on improving the manuscript and for an in-depth review.
Major concerns:
- There are several main pieces in the methods part missing. For example, several aflatoxins are reported in the manuscript, are they all obtained from Sigma-Aldrich? How many total? Are they all use the same detection method (i.e., fluorescence, with the same excitation and emission wavelength)?
All mycotoxin standards were obtained from Sigma-Aldrich (Steinheil, Germany). All data on the determined aflatoxins have been supplemented in the text. All aflatoxins were determined using one method. Appropriate supplementation has been included in the manuscript.
2. What is the Limit of Quantitation (LOQ) for each analyte? How did they determine the LOQ?
The quantification limit for each analyte and the method of its determination have been completed in the text.
3. Clarify the mycotoxin family regarding the toxins they detected, for example, DON and NIV belong to trichothecene, OTA and ZEA belong to OCAs.
The affiliation and origin of the tested toxins has been supplemented in the text.
4. All samples authors used are from those with high risk for chromosomal abnormalities. It is interesting to see and compare the results from normal pregnant women. At least should discuss that in the manuscript in light of the authors' findings that the presence of mycotoxins in those high-risk population.
We used human amniotic fluid samples of high risk pregnancies because it was approved by Ethical Committee. Such samples from normal healthy pregnancies could be collected only if a woman agrees for amniocentesis which is a risky procedure and therefore informed consent is very difficult to obtain. However, samples with normal genetic results may constitute a kind of control subgroup for genetically abnormal samples. It is now added to the discussion as a limitation.
5. Table 1 is one of the main results and is listed in the supplementary materials. Authors should summarize those results for AFs into the Figures with different AFs in the main text to be consistent with other mycotoxins presented in the manuscript. Authors should also list the quantitative results from other mycotoxins in the supplementary materials.
The positive results for aflatoxins are presented in the table (Table 1., Main document) because the graphs was not readable. In addition, all graphs were merged. The results of the quantitative analysis of all mycotoxins are summarized in Table 1 in supplementary materials.
Minor concerns:
- In line 264 the word “microl” has been corrected on “microliters”.
Reviewer 2 Report
The information in lines 96-100 would simplify the readers' understanding if the information was presented as a figure, illustration or scheme.
The figurers 1-3 should be merged. Currently the figures take large space and the information about original sample numbers is most likely not of value for the readers. It would be of value if the "normal/baseline" concentration for each of the toxins could be presented in the figures.
Author Response
The authors thank you very much for valuable tips on improving the manuscript.
- The information in lines 96-100 would simplify the readers' understanding if the information was presented as a figure, illustration or scheme.
The information from the lines 96-100 has been included in the table
2. The figurers 1-3 should be merged. Currently the figures take large space and the information about original sample numbers is most likely not of value for the readers. It would be of value if the "normal/baseline" concentration for each of the toxins could be presented in the figures.
The figures 1-3 have been merged. The use of the original numbering allows the reader to use Table 1 included in the supplementary materials. Unfortunately, there are no standards for the content of mycotoxins in body fluids, therefore it is not possible to present the "normal / baseline" in charts.
Reviewer 3 Report
The authors in this study have conducted analysis to test the presence of mycotoxins in the amniotic fluid of pregnant women and to understand its risk to fetuses.
Previous studies have already shown the presence of mycotoxins such as aflatoxin and ochratoxin A in human amniotic fluids, the current study also detects the same in their set of samples. This is the first report showing NIV and DON toxins in amniotic fluids.
Following are my questions and suggestions to the authors:
- Why is the study only focused on checking if mycotoxins lead to genetic defects in fetus, why other characteristic symptoms of mycotoxicosis was not correlated? Are there any studies that show a direct effect of mycotoxins with genetic abnormalities?
- What are some of the sources of exposure to mycotoxins (each of them) should be elaborated in the introduction.
- In the results section for the samples indicating aflatoxin ,genetic defect is found to be in 19% (6/31) of the samples. How did you come up with 0-22%.
- How many of the samples detecting NIV, gave birth to offspring with genetic defect?
- Why NIV is prevalent more in Poland? What are the different sources? Are all the samples collected from Poland?
- What is the reference for zearalenone not being able to cross over the placenta? What could be some other reasons for this mycotoxin not being detected?
Grammar and references:
- Abstract: Line 11: "of the all' should be 'of all the'
- Aflatoxin and Amniotic fluids have the same acronym AF used in this study. Change one of them.
- Please add references ot the sentence 73: 'Fetus has a unique vulnerability to deleterious health effects resulting from in utero exposure to environmental chemicals, because of rapid cell proliferation, poor detox ification or elimination capacity, high absorption rate, and immature repairing mechanisms'
- Line 92- Please elaborate the meaning of field 'Both toxins that formed in the field'
- Line 168- through- spelling error.
What are the future plans for this study? Since genetic defect is not seen in all samples that show mycotoxins, what are the other risks that could be examined?
Author Response
Reviewers 3
The authors thank you very much for valuable tips on improving the manuscript and for an in-depth review.
- Why is the study only focused on checking if mycotoxins lead to genetic defects in fetus, why other characteristic symptoms of mycotoxicosis was not correlated? Are there any studies that show a direct effect of mycotoxins with genetic abnormalities?
We hypothesized that mycotoxins are present in human amniotic fluid in second trimester of pregnancy. So far due to ethical limitations the only samples available were those from high risk patients for chromosomal abnormalities based on first trimester screening. It is well known that vast majority of such patients are false positive for genetic problems and other pathologies like maternal infections or fetal anemia may be responsible for out of range non-invasive prenatal results. Thus one high risk patient may have genetic problem confirmed in invasive test like amniocentesis while the other may have normal fetal karyotype. We did not correlate any symptoms of mycotoxicosis with genetic results in this study. To our best knowledge we did not find other studies binding mycotoxins with genetic abnormalities in human beings. There are several animal model-based publications available.
2. What are some of the sources of exposure to mycotoxins (each of them) should be elaborated in the introduction.
The sources of mycotoxins exposure were added to the text.
3. In the results section for the samples indicating aflatoxin ,genetic defect is found to be in 19% (6/31) of the samples. How did you come up with 0-22%.
The authors thank you for the insightful review. There is actually a mistake. After making the table, it is easier to count the positive samples (the table is included in the manuscript). There were 27 positive samples, 6 of them were from mothers who gave birth to a child with a genetic defect. The mistake has been corrected.
4. How many of the samples detecting NIV, gave birth to offspring with genetic defect?
NIV was present in 29 tested samples. Of these, four fetuses had genetic defects (13.8%). Missing data has been included in the text
5. Why NIV is prevalent more in Poland? What are the different sources? Are all the samples collected from Poland?
NIV is synthesized by many Fusarium species typical for Poland. In the climatic zone of Poland, F. graminearum, F. culmorum and F. cerealis are one of the most common fungi and they produce nivalenol. They are found mainly in wheat and maize. Due to the common nature of these crops, it is often detected in cereal products, especially in recent years.
6. What is the reference for zearalenone not being able to cross over the placenta? What could be some other reasons for this mycotoxin not being detected?
This is only our hypothesis. It is a very common toxin in all crops and surface waters. So we will be very surprised by its absence in the amniotic fluids. Further extensive research is needed to clearly determine the possibility of zearalenone transfer to the amniotic fluid and the fetus.
7. All grammar and spelling notes have been corrected.
8. What are the future plans for this study? Since genetic defect is not seen in all samples that show mycotoxins, what are the other risks that could be examined?
We plan to extend our study to normal healthy women with low risk for chromosomal abnormalities. Also we intend to search for correlations of mycotoxins with non-genetic congenital malformations such as clefts, spina bifida, diaphragmatic hernia, heart defects, limb abnormalities etc as ell as pregnancy complications like fetal growth restriction, preeclampsia, spontaneous preterm birth, rupture of membranes or gestational diabetes and macrosomia.
Round 2
Reviewer 1 Report
The revision is sufficient.
Author Response
As suggested by the reviewer, the text has been thoroughly checked and typos have been corrected. Additionally, the authors decided to correct Tables 1.